# PRACTICAL MASSIVELY PARALLEL MONTE-CARLO TREE SEARCH APPLIED TO MOLECULAR DESIGN

**Xiufeng Yang**
Chugai Pharmaceutical Co., Ltd
yangxiufengsia@gmail.com

**Tanuj Kr Aasawat**
Parallel Computing Lab - India, Intel Labs
tanuj.aasawat@intel.com

**Kazuki Yoshizoe**[*]
RIKEN Center for Advanced Intelligence Project
yoshizoe@acm.org

## ABSTRACT

It is common practice to use large computational resources to train neural networks, known from many examples, such as reinforcement learning applications. However, while massively parallel computing is often used for training models, it is rarely used to search solutions for combinatorial optimization problems. This paper proposes a novel massively parallel Monte-Carlo Tree Search (MP-MCTS) algorithm that works efficiently for a 1,000 worker scale on a distributed memory environment using multiple compute nodes and applies it to molecular design. This paper is the first work that applies distributed MCTS to a real-world and non-game problem. Existing works on large-scale parallel MCTS show efficient scalability in terms of the number of rollouts up to 100 workers. Still, they suffer from the degradation in the quality of the solutions. MP-MCTS maintains the search quality at a larger scale. By running MP-MCTS on 256 CPU cores for only 10 minutes, we obtained candidate molecules with similar scores to non-parallel MCTS running for 42 hours. Moreover, our results based on parallel MCTS (combined with a simple RNN model) significantly outperform existing state-of-the-art work. Our method is generic and is expected to speed up other applications of MCTS[1].

## 1 INTRODUCTION

A survey paper on MCTS, published in 2012, has cited 240 papers, including many game and non-game applications (Browne et al., 2012). Since the invention of Upper Confidence bound applied to Trees (UCT) (Kocsis & Szepesvári, 2006) (the most representative MCTS algorithm) in 2006, MCTS has shown remarkable performance in various problems. Recently, the successful combination with Deep Neural Networks (DNN) in computer Go by AlphaGo (Silver et al., 2016) has brought MCTS into the spotlight. Combining MCTS and DNN is becoming one of the standard tools for solving decision making or combinatorial optimization problems. Therefore, there is a significant demand for parallel MCTS. However, in contrast to the enormous computing resources invested in training DNN models in many recent studies, MCTS is rarely parallelized at large scale.

Parallelizing MCTS/UCT is notoriously challenging. For example, in UCT, the algorithm follows four steps, selection-expansion-rollout-backpropagation. Non-parallel vanilla UCT updates (backpropagates) the values in the tree nodes after each rollout. The behavior of the subsequent selection steps depends on the results of the previous rollouts-backpropagation. Therefore, there is no apparent parallelism in the algorithm.

---

[*]This work was done while all the authors were at RIKEN.
[1]Code is available at https://github.com/yoshizoe/mp-chemts

Using *virtual-loss* technique (explained in section 2.3), MCTS has been efficiently parallelized on shared-memory single machine environment, where the number of CPU cores are limited (Chaslot et al., 2008; Enzenberger & Müller, 2010; Segal, 2010). However, there is limited research on large-scale parallel MCTS using distributed memory environments. Only two approaches scale efficiently on distributed memory environment, but these were only validated in terms of the number of rollouts and the actual improvement is not validated (Yoshizoe et al., 2011; Graf et al., 2011).

Recently, the combination of (non-parallel) MCTS and DNN has been applied to molecular design problems, which aims to find new chemical compounds with desired properties (Yang et al., 2017; Sumita et al., 2018), utilizing the ability of MCTS to solve single-agent problems. In general, designing novel molecules can be formulated as a combinatorial optimization or planning problem to find the optimal solutions in vast chemical space (of $10^{23}$ to $10^{60}$, Polishchuk et al. (2013)) and can be tackled with the combinations of deep generative models and search (Kusner et al., 2017; Gómez-Bombarelli et al., 2018; Jin et al., 2018; Popova et al., 2018; 2019; Yang et al., 2017). However, there are no previous studies about massively parallel MCTS for molecular design.

In this paper, we propose a novel distributed parallel MCTS and apply it to the molecule design problem. This is the first work to explore viability of distributed parallel MCTS in molecular design. Our experimental results show that a simple RNN model combined with massively parallel MCTS outperforms existing work using more complex models combined with Bayesian Optimization or Reinforcement Learning (other than UCT).

## 2 BACKGROUND

### 2.1 (NON-PARALLEL) MCTS

In 2006, Kocsis and Szepesvári proposed UCT based on a Multi-Armed Bandit algorithm UCB1 (Auer et al., 2002), which is the first MCTS algorithm having a proof of convergence to the optimal solution. It has shown good performance for many problems, including the game of Go (Gelly et al., 2006).

One round of UCT consists of four steps, as shown in Fig. 1. It repeats the rounds for a given number of times or until a specified time has elapsed.
**Selection:** The algorithm starts from the root node and selects the child with the highest UCB1 value (Fig. 2 left) until it reaches a leaf node. For each child $i$, $v_i$ is the number of visits, $w_i$ is the cumulative reward, and $V$ is the number of visits at the parent node. Exploration constant $C$ controls the behavior of UCT: the smaller, the more selective; the greater, the more explorative.
**Expansion:** If the number of visits exceeds a given threshold, expand the leaf node (add the children of the leaf to the tree) and select one of the new children for simulation. Do nothing otherwise.
**Simulation:** UCT then evaluates the node by a *simulation*. This step is often called playout or rollout. A simple example of rollout is to go down the tree by selecting a random child at each node until it reaches a terminal node and returns the value at the terminal node as the reward $r$ (win or loss for games). Replacing rollout with a DNN based evaluation is becoming more popular following the success of AlphaGo.

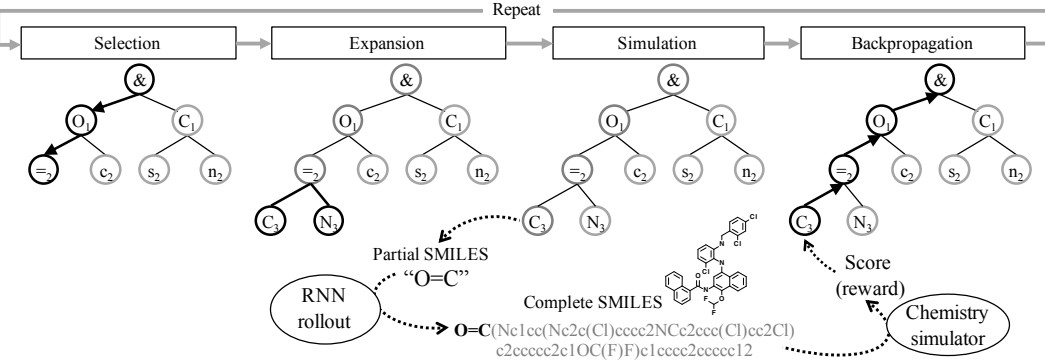

Figure 1: Four steps of (non-parallel) MCTS, with simulation for molecular design.

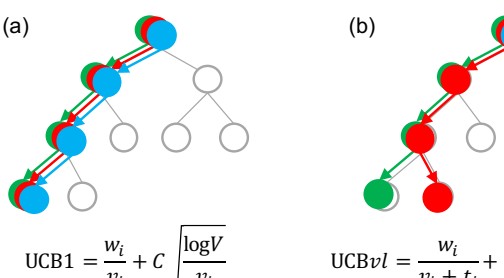

(a)

(b)

Green worker goes first, increments $t_i$ on the path and reaches a leaf.

Red goes second, increments $t_i$ and avoids the leaf because of the penalty.

Blue further avoids the same path because of the greater penalty.

$$\text{UCB1} = \frac{w_i}{v_i} + C\sqrt{\frac{\log V}{v_i}}$$

$$\text{UCB}vl = \frac{w_i}{v_i + t_i} + C\sqrt{\frac{\log(V + T)}{v_i + t_i}}$$

Figure 2: (a) parallel UCT using UCB1 (failed) (b) parallel UCT with *virtual loss*, and the search paths of three parallel workers shown in solid circles, (green, red, and blue, from left to right).

**Backpropagation:** UCT finally traverses the path all the way back to the root, and updates the values of the nodes in the path ($w_i = w_i + r$, $v_i = v_i + 1$).

## 2.2 SOLVING MOLECULAR DESIGN USING MCTS

The Simplified Molecular-Input Line-Entry System (SMILES) (Weininger, 1988) is a standard notation in chemistry for describing molecules using ASCII strings, defined by a grammar. SMILES uses ASCII symbols to denote atoms, bonds, or structural information such as rings. For example, SMILES for Carbon dioxide is "O=C=O" where "=" means a double bond, and for Benzene, it is "C1=CC=CC=C1" or "c1ccccc1" which forms a ring by connecting the two "C1"s. The search space for molecular design can be defined as a tree based on the SMILES, where the root node is the starting symbol (denoted as "&", usually not shown) and each path from the root to a depth $d$ node represents a left-to-right enumeration of the first $d$ symbols in the (partial) SMILES string (see Fig. 1. The subscripts shows the depth.).

Yang et al. (2017) were the first to apply non-parallel MCTS to this problem using the AlphaGo-like approach, which combines MCTS with DNN and a computational chemistry simulator. They trained an RNN model with a chemical compound database, and the RNN predicts the next symbol of a partial SMILES string which is given as the input. The model is used for expansion and rollout.

**Expansion for Molecular Design:** The node ($=_2$) in Fig. 1 denotes SMILES strings starting with "O=". The RNN receives "O=" as input and outputs the probability of the next SMILES symbols. Instead of simply adding all symbols as child nodes, the low probability branches are pruned based on the output of the model.

**Simulation for Molecular Design:** Fig. 1 illustrates the simulation step for molecular design. Node ($C_3$) denotes SMILES strings starting with "O=C". Firstly, a rollout generates a complete SMILES string, a new candidate molecule, by repeatedly sampling the next symbol using the model, until the model outputs the terminal symbol. Then, a computational chemistry simulator receives the SMILES string and calculates the target chemical property for the molecule, to determine the reward.

## 2.3 PARALLEL MCTS

Non-parallel MCTS finds the most promising leaf one at a time in the selection step. In parallel MCTS, multiple workers must find the promising leaves (to launch simulations from) in parallel, hence, should find leaves speculatively without knowing the latest results.

Fig. 2 shows the example of a three *workers* case (a worker means a thread or a process, which runs on either a CPU or GPU). If the workers (shown in green, red, blue) follow the UCB1 formula, all end up at the same leaf node (Fig. 2-(a)), and the parallelization fails.

The *Virtual loss* (Chaslot et al., 2008) let the workers find the promising but different leaves. The original UCB1 is modified to UCB$vl$ shown in Fig. 2-(b), where $t_i$ is the number of workers currently searching in the subtree of the child node $i$, and $T$ is the sum of $t_i$ of the children. It is called *virtual loss* because it assumes that the current ongoing simulations will obtain 0 reward. With this modification, UCB$vl$ value is penalized based on the number of workers, which makes the subsequent workers avoid the subtrees already being explored (see Fig. 2-(b)).

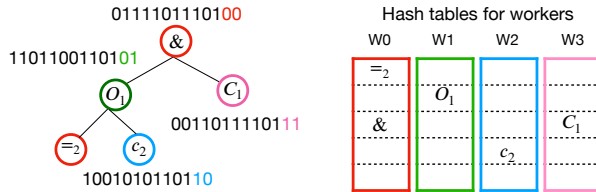

Figure 3: Hash-driven parallel search distributes the nodes to workers based on a hash function. The *home worker* of the nodes are defined by the trailing 2 bits.

Parallel UCT with virtual loss was proved to improve the strength of shared-memory (using up to 16 cores) Go programs (Chaslot et al., 2008; Enzenberger & Müller, 2010). Segal's experiment on an emulated multithreaded machine that assumes no communication overhead shows speedup on up to 512 threads. However, his experiments on real distributed machines did not provide significant speedup beyond 32 cores (Segal, 2010).

## 2.4 Hash-driven Parallel Search and Uniform Load Balancing

When many workers perform a search in parallel, it is essential to distribute the search space as evenly as possible. If there is an imbalance in the workload, overall performance will be less efficient because only some workers will continue the computation while the rest remain idle. If all the search space is explicitly given in advance, this is a trivial matter. However, in case of problems such as games and combinatorial optimization, where the search space is generated while the search is performed, it is difficult to distribute the search space evenly.

Hash-driven (HD) parallel search is one of the methods for resolving this difficulty; Evett et al. (1995) (PRA*) and Romein et al. (1999) (TDS: Transposition-table Driven Scheduling) developed this method independently and applied to parallel Iterative Deepening A* search. Kishimoto et al. (2013) later applied it to parallel A* search. HD parallel search requires a hash function that defines the hash key of each node. Each time the algorithm creates a node during the search, it assigns the node to a specific worker, the *home worker* (or the *home processor)*, based on the hash function, and achieves a near-equal load balancing if the hash function is sufficiently random (such as Zobrist Hashing (Zobrist, 1970)).

Figure 3 illustrates this method. The hash function randomly divides the tree into four partitions (shown in four colors), and each worker holds the nodes in the assigned partition in its hash table. A distinct drawback of this method is that it requires communication between workers almost every time the algorithm traverses a branch (unless the two nodes are on the same worker). The key to efficient parallel speedup is the trade-off between uniform load balancing and frequent communication. The experimental results for IDA* and A* (Evett et al., 1995; Romein et al., 1999; Kishimoto et al., 2013) prove its efficiency for these search algorithms. However, a straightforward application of HD parallel search to UCT is not efficient enough as described below.

## 3 Distributed Parallel MCTS and Molecular Design

### 3.1 TDS-UCT: Hash-driven Parallel MCTS

Here we describe the application of Hash-driven parallel search to UCT, TDS-UCT. Figure 4-(a) illustrates the behavior of the four steps in TDS-UCT.

**Selection:** The worker which holds the root node selects the best child ($O_1$ in Fig. 4) based on UCB*vl* formula (Fig. 2-(b)). It then sends a *selection message*, which holds information of $O_1$, to the *home worker* of $O_1$ (the green worker in Fig.4). If a worker receives a *selection message*, it selects the best child of the node and pass the message to another worker until the message reaches a leaf node. The worker-count $t_i$ of each node is incremented during the step.
**Expansion:** If a worker receives a *selection message* and the node is a leaf, the expansion step is done in the same way as in non-parallel MCTS.
**Simulation:** Simulation is done by the *home worker* of the leaf (the green worker in Fig. 4) in the

same way as in non-parallel MCTS.

**Backpropagation:** After a simulation (at $C_3$), a *backpropagation message*, which holds the reward $r$, is sent to the parent ($=_2$). The workers pass the *backpropagation message* to the parent until it reaches the root node. At each node, the values are updated by the corresponding worker ($w_i = w_i + r$, $v_i = v_i + 1$, $t_i = t_i - 1$).

To reduce the the number of idle workers, the sum of the number of *selection messages*, *backpropagation messages* and ongoing simulations must be more than the number of workers. Yoshizoe et al. (2011) and Graf et al. (2011) independently proposed to control the number to $N \times \#workers$ where $N$ is the *overload factor*. We used $N = 3$ following the experimental results in the two papers.

### 3.2 TDS-DF-UCT: DEPTH-FIRST REFORMULATION OF TDS-UCT

Scalability of TDS-UCT is limited because of the communication contention around the root node. In TDS-UCT, all backpropagation messages are sent up until to the root node. As the number of messages increase, the workers that hold shallow nodes (especially the root node) spend more time for communication. Because of this problem, the scalability of TDS-UCT quickly diminishes beyond 100 workers (Yoshizoe et al., 2011).

For solving the communication contention, TDS-df-UCT was proposed based on the observation that the promising part of the tree does not change so often after each simulation (Yoshizoe et al., 2011). The next selection step will likely reach a node that is close to the previously selected leaf node, as shown in Fig. 4-(a). The leaf is $C_3$ for the 1st selection step and $N_3$ for the 2nd selection step. Each job message in TDS-df-UCT carries a history table which contains the history of the siblings of the nodes in the current path. After each simulation, TDS-df-UCT updates the values in the node and the history table, and re-calculate the UCB$vl$ of the siblings. Unlike TDS-UCT, TDS-df-UCT will backpropagate only if the UCB$vl$ value of the nodes in the current path is exceeded by one of the other siblings in the history table.

Fig. 4-(b) shows a trajectory of a message in TDS-df-UCT. After the first selection-expansion-simulation, it does not send the backpropagation message to the parent of $=_2$. Instead, it directly sends a *selection message* to $N_3$, as $=_2$ is still the most promising node. This technique dramatically reduces the number of messages sent to the root node by staying at the more promising part, thus solving the communication contention problem.

### 3.3 MP-MCTS: ADDRESSING SHORTCOMINGS OF TDS-DF-UCT

TDS-df-UCT carries the history only in the messages, hence, requires more time to propagate new results to other jobs. As a result, TDS-df-UCT skips backpropagations too often because it overlooks that other workers may have found more promising part of the tree, therefore making the tree wider and shallower. In our MP-MCTS, along with its own statistical information ($w$, $v$, $t$), each node maintains the history table as shown in Fig. 4 (c). It accelerates the propagation of the latest information at the cost of extra (minor) memory usage (see Appendix A.2 for the memory usage analysis). This modification to TDS-df-UCT is crucial for improving the behavior of the parallel MCTS to generate a deeper tree. We illustrate in the experimental results that the quality of the

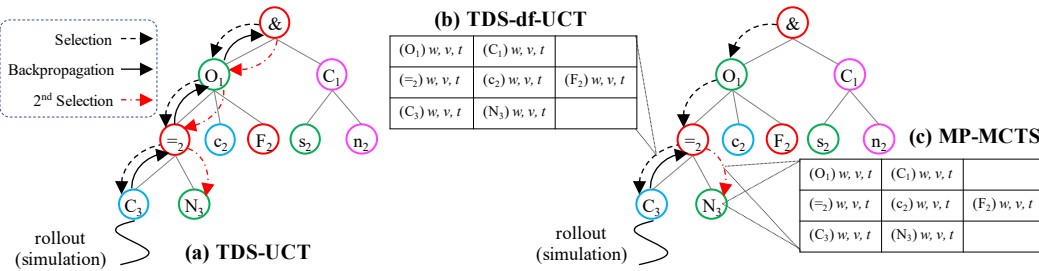

Figure 4: The trajectory of a message in (a) TDS-UCT (b) TDS-df-UCT and (c) MP-MCTS. TDS-df-UCT has the history table only in the messages and MP-MCTS stores it in the nodes also.

solution and the average depth of the tree generated by $N$-parallel MP-MCTS are both similar to the ideal case obtained by running the non-parallel MCTS for $N$-times more wall-clock time.

## 4 EXPERIMENT METHODOLOGY

To evaluate the quality of the solutions and to compare against state-of-the-art methods, we use the octanol-water partition coefficient (logP) penalized by the synthetic accessibility (SA) and large Ring Penalty score, a popular benchmarking physicochemical property (Kusner et al., 2017; Jin et al., 2018; You et al., 2018a; Popova et al., 2019; Maziarka et al., 2020) used in molecular design.

**Model and Dataset:** We use the GRU-based model and pre-trained weights publicly available on the repository of ChemTS (Yang et al., 2017), which mainly consists of two stacked GRU units. Input data represents SMILES symbols using 64-dim one-hot vectors. The first GRU layer has 64-dim input/256-dim output. The second GRU layer has 256-dim input/256-dim output, connected to the last dense layer, which outputs 64 values with softmax. The model was pre-trained using a molecule dataset that contains 250K drug molecules extracted from the ZINC database, following (Kusner et al., 2017; Yang et al., 2017; Jin et al., 2018).

**MCTS implementation:** The GRU model explained above takes a partial SMILES string as the input and outputs the probability of the next symbol. We use this model for both Expansion and Simulation. In the expansion step, we add branches (e.g., SMILES symbols) with high probability until the cumulative probability reaches 0.95. For a rollout in the simulation step, we repeatedly sample over the model to generate a complete SMILES string. Then the string is passed to a computational chemistry tool, RDKit (Landrum et al., 2006) for calculating the penalized logP score, which is commonly used in existing work. We use the reward definition described in Yang et al. (Yang et al., 2017) which is normalized to $[-1, 1]$ and consider the same value for exploration constant, $C = 1$.

**Other experimental settings:** Algorithms are implemented using Keras with TensorFlow and MPI library for Python (mpi4py). All experiments, unless otherwise specified, were run for 10 minutes on up to 1024 cores of a CPU cluster (each node equipped with two Intel Xeon Gold 6148 CPU (2.4GHz, 20 cores) and 384GB of memory), and one MPI process (called worker in this paper) is assigned to one core.

## 5 EXPERIMENT RESULTS

To evaluate the distributed MCTS approaches, we study the quality of the solutions obtained, analyze the performance of different parallel MCTS algorithms, and compare the performance of MP-MCTS with state-of-the-art work in molecular design.

**Maximizing penalized logP score.** Table 1 presents the penalized logP score of distributed MCTS approaches for varying number of CPU cores. With increasing number of cores available, more number of simulations can be performed in parallel, which improves the quality of the score. We performed 10 runs for each settings, and show the average and standard deviation of the best scores. TDS-UCT suffers from communication contention, with increasing number of cores the load imbalance becomes more significant, hence yields lower score. TDS-df-UCT achieves more uniform load balancing but only slightly improves the score (discussed later in this section). Our MP-MCTS, which mitigates the issues of other approaches, shows strict improvement in score with increase in number of cores leveraged.

**Quality of parallel solution over non-parallel solution.** As mentioned earlier, any parallel MCTS must speculatively start to search before knowing the latest search results, and it may return different outcomes from those of the non-parallel version. In addition, the exploration and exploitation trade-off of distributed MCTS is controlled via the virtual-loss based UCB$vl$ instead of theoretically guaranteed UCB1 (Browne et al., 2012). Hence, it is significant to compare the quality of distributed MCTS solution with non-parallel MCTS (cf. strength speedup (Appendix A.1)).

The bottom row of Table 1 presents the penalized logP score for non-parallel MCTS. Note that non-parallel MCTS was run for equivalent core-hours (for example, 256 cores for non-parallel MCTS indicates it was run for $256 \times 10$ minutes on a single core; while distributed MCTS is run on 256 cores

Table 1: Penalized logP score (higher the better) of TDS-UCT, TDS-df-UCT and MP-MCTS, with 10 minutes time limit, averaged over 10 runs.
⋆ 4, 16, 64 and 256 cores for non-parallel-MCTS indicates the algorithm was performed for 40 (4×10), 160 (16×10), 640 (64×10), and 2560 (256×10) minutes. For 1024×10 minutes, non-parallel-MCTS was not performed due to the huge execution time (∼170 hours).

| cores
Methods | 4 | 16 | 64 | 256 | 1024 |
|---|---|---|---|---|---|
| TDS-UCT | 5.83±0.31 | 6.24±0.59 | 7.47±0.72 | 7.39±0.92 | 6.22±0.27 |
| TDS-df-UCT | 7.26±0.49 | 8.14±0.34 | 8.59±0.49 | 8.22±0.41 | 8.34±0.46 |
| **MP-MCTS** | 6.82±0.76 | 8.01±0.61 | 9.03±0.85 | **11.46±1.52** | **11.94±2.03** |
| ⋆non-parallel-MCTS
(#cores × 10 minutes) | 6.97 ± 0.49 | 8.54 ± 0.34 | 9.23 ± 0.53 | 11.17 ± 0.88 | – |

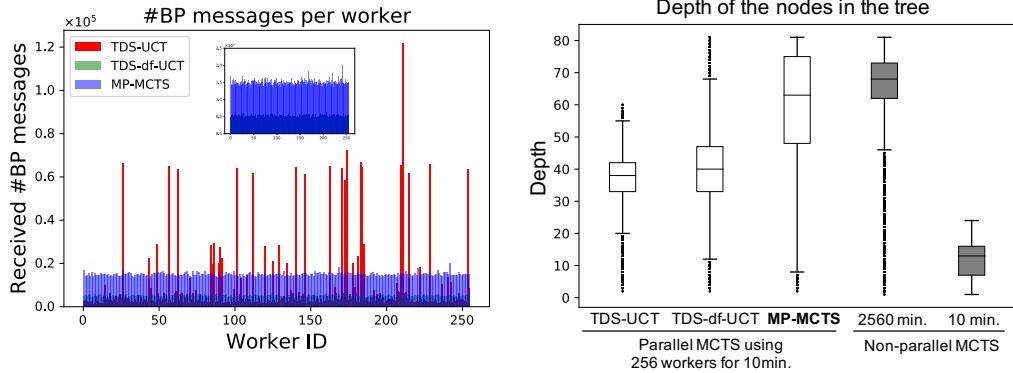

Figure 5: (a) Number of back-propagation (BP) messages received by each of 256 workers. (b) Box-whisker plot showing the depth of the search tree for different parallel MCTS variants.

for 10 minutes). While taking much less time, the distributed-MCTS is on-par and yields higher score than non-parallel when large computing resources (i.e. with 256 and 1024 cores) are leveraged.

**Message traffic reduction at root.** Fig. 5 (a) presents back-propagation messages received by each worker. MP-MCTS and TDS-df-UCT lead to better *load balance* across all workers, while for TDS-UCT, root's and few other hot nodes' home worker get overwhelmed by the messages, and suffers from extremely high load-imbalance. Note that TDS-df-UCT generates a shallower tree, resulting in fewer BP messages.

**Depth of the search tree.** Fig. 5 (b) compares the depth of the nodes in the search tree generated by parallel and non-parallel MCTS. TDS-UCT creates shallower tree than the others because of the load imbalance. TDS-df-UCT generates a deeper tree than TDS-UCT but still much shallower than ideal case. It is presumed that TDS-df-UCT does too few backpropagations (see 3.2). MP-MCTS generates a significantly deeper tree by using 256 cores for 10 minutes, which is close to the ideal case of non-parallel MCTS using 2,560 minutes (approx. 42h).

**Comparison against related work in molecular design.** Table 2 presents the top 3 penalized logP scores obtained by the existing state-of-the-art work (description in section 6). The scores of MP-MCTS are the best among 10 runs (10 minutes each), which outperforms the existing work significantly in maximizing the penalized logP score. It is also notable that MP-MCTS significantly improved the score of the GRU-based model (Yang et al., 2017). The bottom two lines compare the results obtained from 10 minutes random sampling from the GRU-based model with score obtained by MP-MCTS which uses the same model (as mentioned in Section 4). *This result suggests the possibility of improving existing work by combining their models with parallel MCTS.*

Table 2: Comparison of the best three penalized logP scores

| Methods | 1st | 2rd | 3rd | |
|---|---|---|---|---|
| JT-VAE (Jin et al., 2018) | 5.30 | 4.93 | 4.49 | reported results |
| GCPN (You et al., 2018a) | 7.98 | 7.85 | 7.80 | reported results |
| MolecularRNN (Popova et al., 2019) | 10.34 | 10.19 | 10.14 | reported results |
| MolDQN (Zhou et al., 2019) | 9.01 | 9.01 | 8.99 | reported results[a] |
| Mol-CycleGAN (Maziarka et al., 2020) | 9.76 | 7.29 | 7.27 | reported results |
| GRU-based (Yang et al., 2017) | 6.47 | 5.65 | 5.01 | 8 hours x 10 runs |
| **MP-MCTS** using GRU | **15.13** | **14.77** | **14.48** | 10 min. x 10 runs |

[a]These scores are based on the updated values in the authors' correction of the paper.

## 6 RELATED WORK

### 6.1 MOLECULAR DESIGN

In molecular design, it is a common approach to use deep generative models (to generate candidate molecules), followed by optimization algorithms to focus on the promising candidates having desired molecular property (mainly Bayesian Optimization (BO) or Reinforcement learning (RL)). Gomez-Bombarelli et al. (Gómez-Bombarelli et al., 2018) were the first to employ variational autoencoders (VAE). Kusner et al. (Kusner et al., 2017) enhanced it to grammar variational autoencoder (GVAE) by combining context free grammars. Both of the above used BO for optimization. Segler et al. (Segler et al., 2018a) focused on generating molecules using LSTM (Hochreiter & Schmidhuber, 1997). Olivecrona et al. (Olivecrona et al., 2017) used simple RNN and Popova et al. (Popova et al., 2018) used GRU for generation, both combined with RL. These work use SMILES representations. Graph-based molecule generation, such as JT-VAE (Jin et al., 2018) and GCPN (You et al., 2018a) generate molecules by directly operating on molecular graphs, optimized with BO and RL respectively. Popova et al. (Popova et al., 2019) proposed MolecularRNN, an improved RNN model by extending the GraphRNN (You et al., 2018b) with RL. MolDQN (Zhou et al., 2019) is based on DQN with three types of actions, Atom addition, Bond addition, and Bond removal.

The above mentioned work do not use MCTS. Jin et al. (2020) applied MCTS for molecule design with multi-objective constraints, where MCTS is first used for extracting important substructures, then a graph-based molecule generation is trained on a data set having these substructures, and outperforms RL based methods. Yang et al. (2017) combined non-parallel MCTS with a simple GRU and outperformed BO based methods in penalized logP. Sumita et al. (2018) later applied the same approach to a wavelength problem using a quantum mechanical simulation method (DFT).

### 6.2 PARALLEL MCTS

Among the work combining DNN and MCTS beyond single-machine, Distributed AlphaGo (using 1202 GPUs) is the only massively parallel work (applied to Go program). However, it has limited scalability because only one machine holds the search tree, and the GPUs in other machines are used only for neural network computation.

Hash driven parallel search was first applied to distributed MCTS by TDS-df-UCT (Yoshizoe et al., 2011) (for an artificial game) and UCT-tree-split (Graf et al., 2011) (for the game of Go) independently, and solves communication contention problem. UCT-tree-split preserves a replicated subtree for each worker which contains the nodes with large number of visits. It periodically synchronizes the subtrees, hence, suffers from higher communication overhead than TDS-df-UCT.

Other distributed parallel MCTS techniques include Root parallelization and Leaf parallelization. *Root parallelization* performs multiple runs of MCTS with different random seeds (Chaslot et al., 2008) and synchronizes the shallow part of the tree. The strength improvement is limited because the size of the tree remains similar to that of non-parallel MCTS (Soejima et al., 2010). Another known parallel method is *Leaf parallelization* (Cazenave & Jouandeau, 2007), but its performance was lower than that of root parallelization.

Among shared-memory environment (having only a few tens of workers), the first reported parallel UCT (Gelly et al., 2006), did not rely on *virtual loss* and it was less efficient. Many other work on shared memory environment use *virtual loss* (Chaslot et al., 2008; Enzenberger & Müller, 2010; Segal, 2010; Liu et al., 2020). However, they are limited to a maximum of 32 workers.

## 7 CONCLUSION

Applying MCTS to molecular design is relatively less explored. Ours is the first work to explore distributed parallel MCTS for molecular design. The extensive experiments have shown that an efficient distributed MCTS significantly outperforms other approaches that use more complex DNN models combined with optimizations such as Bayesian Optimization or Reinforcement Learning (other than UCT). Further, it does not trade off search ability (w.r.t non-parallel MCTS) for a real world problem.

It would be an interesting future work to further enhance the performance of molecular design by using more complex and improved models with parallel MCTS. Also, MP-MCTS could be applied to other MCTS applications, including retrosynthetic analysis to which an AlphaGo-like approach is applied (Segler et al., 2018b). Furthermore, although we analyzed the performance for molecular design, the parallelization technique is independent of the chemistry specific components and could be applied to other game and non-game applications. Our experiments strongly suggest that MCTS can be a better alternative for real-world optimization problems.

### ACKNOWLEDGMENTS

This work is partially supported by JSPS Kakenhi (20H04251). Experiments were conducted on the the RAIDEN computer system at RIKEN AIP.

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

# A    OTHER DETAILS OF PARALLEL MCTS

## A.1    STRENGTH SPEEDUP

It is quite misleading to compare only the number of rollout for evaluating the parallel MCTS algorithms. For example, it is easy to achieve a linear speedup in terms of the number of rollouts using the simplistic root parallel MCTS. However, it is not meaningful because it does not contribute to the actual strength.

For example, the only massively parallel Go program based on DNN and MCTS, distributed AlphaGo, wins 77% of the games against single-machine AlphaGo using approximately 150-fold more resources (1202 GPUs vs. 8 GPUs) (Silver et al., 2016). Distributed AlphaGo is presumed to have limited scalability because only one machine holds the search tree, and the GPUs in other machines are used only for neural network computation. But how efficient is this in reality? How should we estimate the effective speedup of the distributed version?

In the game-AI domain, *strength speedup* is used to evaluate the scalability. First, measure the strength improvement of the non-parallel program, by using $N$-times more thinking time than the default settings. Then, if the parallel version reaches the same strength within the same default thinking time, using $N$-times more computing resources, the speedup is ideal.

Strength speedup was never measured for distributed AlphaGo, but we can guess from the results of other similarly strong Go programs. Based on the self-play experiments of ELF OpenGo (Fig. 7, Tian et al. (2019)), it is enough to use four times more thinking time to achieve 77% win rate against itself. Therefore, at least for ELF OpenGo, 77% win rate means less than or equals to 4-fold strength speedup. Based on this estimation, we presume that distributed AlphaGo is not efficient. (It is also guessed that distributed AlphaGo was designed to be robust against hardware failures and the scalability was not the first priority.)

It is difficult to directly compare Go and molecular design, but our "strength speedup" shows almost ideal speedup, obtaining better results (within error range) than non-parallel MCTS (using $256\times$ more time) by using 256 cores in parallel.

## A.2    EXTRA MEMORY USAGE OF MP-MCTS

In MP-MCTS, the additional memory consumption per worker of storing $w, v, t$ of siblings at depth $d$ is: $C \times d \times b \times n \times t$, where $C = 24$ is a constant representing the memory usage of storing $w, v, t$ (each costs 8 bytes) in a node, $d$ is the depth of the tree, $b$ is the average branching factor, $n$ is the number of stored nodes per worker per second, and $t$ is the execution time(s). In our molecular design case, $b \approx 2.7$, $n \approx 1.8$ (one evaluation on a node costs 0.5s, on average). Assuming $d \approx 20$, the additional memory usage is 2.3KB per worker per second. For other problems (e.g., AlphaGO), presuming $b \approx 9$, $d \approx 20$, and $n \approx 200$ (approximated by one evaluation of a node using DNN costs 5ms), then the additional memory usage is 843KB per worker per second, which is acceptable for recent servers having 10+GB memory per core (or worker) (allows more than 10,000 seconds of thinking time). In case the memory usage increases because of the larger $b$, we can save memory by recording only the top-$K$ branches and ignore less promising branches.

# B    VIRTUAL LOSS DETAILS

## B.1    THE ORIGIN OF VIRTUAL LOSS

One of the first detailed paper about parallel MCTS (Chaslot et al. 2008 (Chaslot et al., 2008)) already describes the use of virtual loss (page 5, section 3.3 bottom paragraph titled "Virtual loss"). The explained behavior about avoiding multiple threads visiting the same leaf is the same as we described in Section 2.3 of the main text. Yoshizoe et al. 2011 (Yoshizoe et al., 2011) explains virtual loss in more detail.

Chaslot et al. mentions that Coulom[2] suggested, in personal communication, to assign one virtual loss per one thread during phase 1 (e.g. the selection step). However, when we personally contacted Coulom to confirm the inventor of virtual loss, it turned out that Coulom was not the inventor. We could not track further and the original inventor of virtual loss is still unknown.

Recently, in early 2020, Liu et al. proposed a method very similar to the vanilla virtual loss (eq. 1) named Watched the Unobserved in UCT (WU-UCT) (Liu et al., 2020), (eq. 2), We presume Liu et al. misunderstood or overlooked the explanation of virtual loss formula in existing work for some reasons (although they refer to Chaslot et al. (Chaslot et al., 2008), mentioned above). The explanation about virtual loss in the related work section in (Liu et al. (Liu et al., 2020)) and also their comment at the OpenReview website[3] are wrong. In Response to reviewer #4 (part 1 of 2) they say,

> "To our best knowledge, NONE of the existing TreeP algorithms (or any existing parallel MCTS algorithm) updates the visit counts BEFORE the simulation step finishes. TreeP only updates the values ahead of time using virtual loss. This is also the case for the work [1] and [2]."

However, this is not true. To our best knowledge, most of the existing work on multithreaded parallel MCTS published after 2008 updates the values BEFORE the simulation step completes. (Actually, we can not find the benefit of adding virtual loss AFTER the simulation step.)

Liu et al. say AlphaGo (referred as [1] and [2] in their comment) do not update virtual loss BEFORE the simulation step completes. However this is presumed to be a misunderstanding caused by the explanation in the first AlphaGo paper (Silver et al., 2016). The paper explains that virtual loss value is added during the Backup phase, so it sounds like the value is added after the simulation (explained in Methods, Search Algorithms Section, in the paper (Silver et al., 2016)). However, if you read carefully, the paper says that the virtual loss is added before the end of the simulation, and removed (subtracted) after the end of the simulation. Therefore, AlphaGo does update the visit count before the simulation step finishes. The main difference between WU-UCT and existing Virtual-loss-based parallel MCTS is the difference of the two formulas (eq. 1 and 2).

$$\text{UCB}vl = \frac{w_i + 0}{v_i + t_i} + C\sqrt{\frac{\log(V + T)}{v_i + t_i}} \tag{1}$$

$$\text{UCB}wu = \frac{w_i}{v_i} + C\sqrt{\frac{\log(V + T)}{v_i + t_i}} \tag{2}$$

$$\text{UCB}vl_{LCB} = \frac{w_i + \text{LCB}}{v_i + t_i} + C\sqrt{\frac{\log(V + T)}{v_i + t_i}}, \ \text{LCB} = \min\left\{0, t_i\left(\frac{w_i}{v_i} - C\sqrt{\frac{\log(V + T)}{v_i + t_i}}\right)\right\} \tag{3}$$

## B.2 COMPARISON OF VIRTUAL LOSS FORMULAS

We compared the results of our MP-MCTS using three different virtual loss formulas, the vanilla virtual loss, WU (Liu et al., 2020), and a new formula shown in eq. 3, $\text{UCB}vl_{LCB}$ (LCB stands for Lower Confidence Bound). Vanilla virtual loss assumes zero reward from the ongoing simulations and WU assumes the reward remains unchanged. $\text{UCB}vl_{LCB}$ assumes something between these two, assumes a decreased reward estimated by Lower Confidence Bound.

Table 3 shows the logP score results of our MP-MCTS using three virtual loss formulas, averaged for 10 runs. The results suggests that, for molecular design, WU and $\text{UCB}vl_{LCB}$ do not improve the results over vanilla virtual loss.

---

[2] Remi Coulom, invented the first MCTS algorithm (Coulom, 2006) and applied to his Go program CrazyStone (before UCT).

[3] https://openreview.net/forum?id=BJlQtJSKDB

Table 3: Penalized logP score obtained by MP-MCTS using different virtual loss formulas for 256 and 1024 workers.

| Methods | 256 | 1024 |
|---|---|---|
| UCB$wu$ (Liu et al., 2020) | $10.19 \pm 0.82$ | $10.37 \pm 1.01$ |
| UCB$vl_{LCB}$ | $10.75 \pm 1.32$ | $10.78 \pm 0.27$ |
| **UCBvl (vanilla)** | $11.46 \pm 1.52$ | $11.94 \pm 2.03$ |

### B.3 EXAMPLES OF VIRTUAL LOSS IN CODES

It is common to use virtual loss for parallel game programs. We show real examples of virtual loss implementations in existing open source Go or other game programs, both before and after AlphaGo. We can see the examples of the usage of virtual loss, and in all of these, the values are updated BEFORE the simulation step completes. It is also interesting to note that the majority of these work modify virtual loss equation for their implementations because they rely on different variations of UCT, such as P-UCT (Rosin, 2010).

**Fuego:** Fuego (Enzenberger et al., 2010), one of the best open source Go programs until 2015, is a generic game library. It started to use virtual loss in 2008 (from r677, committed on Nov. 27, 2008). In the `PlayGame` function starting from line 673 (in the following URL), it calls `AddVirtualLoss` at line 685, clearly before the playout (simulation), playouts start right below at line 693 after `StartPlayouts`.

```
https://sourceforge.net/p/fuego/code/HEAD/tree/tags/VERSION_0_3/
smartgame/SgUctSearch.cpp
```

**ELF OpenGo:** It is a Go program and generic game library developed by FaceBook researchers (Tian et al., 2019). Source code of the search part is in the following URL. In the `single_rollout` function starting from line 258, it calls `addVirtualLoss` at line 282. (Please note that here "rollout" means the one whole cycle of the UCT, start selection from the root, reach a leaf, do a (random) rollout, and backpropagate.)

```
https://github.com/pytorch/ELF/blob/113aba73ec0bc9d60bdb00b3c439bc60fecabc89/
src_cpp/elf/ai/tree_search/tree_search.h
```

**LeelaZero:** An open source Go program. In `play_simulation` function starting from line 59, `virtual_loss` updates the values Please note that this is a part of the selection step. `play_simulation` is recursively called at line 88 or 94.

```
https://github.com/leela-zero/leela-zero/blob/0d1791e3f4de1f52389fe41d341484f4f66ea1e9/
src/UCTSearch.cpp
```

**AQ:** An open source Go program. Virtual loss is added before simulations at line 142, and subtracted at line 199 or around line 247 after simulations. Also it is interesting to note that AQ uses virtual loss in two different ways, one for random rollouts and one for Neural Network based evaluation.

```
https://github.com/ymgaq/AQ/blob/36f6728f2f817c2fb0c69d73b00ce155582edb10/
src/search.cc
```

## C TOP MOLECULES

### C.1 LOGP OPTIMIZATION

Figure 1 (a) shows the top 3 molecules by MP-MCTS for penalized logP optimization. MP-MCTS can design molecules with extremely high penalized logP scores, which demonstrates that our MP-MCTS algorithm has the ability of identifying the promising branches. However, these molecules are not desirable because they are likely to have inaccurate predicted properties, which shows an limitation of maximizing penalized logP using an empirical tool (e.g. RDKit).

Therefore it would be interesting to apply MCTS based approach for a different optimization problem with more accurate simulations. Figure 1 (b) shows the best three molecules designed by MP-MCTS for another problem, wavelength property optimization (explained below).

Table 4: Wavelength (nm) score (higher the better) with 6 hours time limit, averaged over 3 runs. $-$ indicates that experiments under the settings were not performed.
$\star$ 4 and 16 cores for non-parallel-MCTS indicates the algorithm was performed for 4$\times$6 and 16$\times$6 hours. For 64$\times$6, 256$\times$6 and 1024$\times$10 hours, non-parallel-MCTS was not performed due to the huge execution time.

| cores
Methods | 4 | 16 | 64 | 256 | 1024 |
|---|---|---|---|---|---|
| MP-MCTS | 1038.7$\pm$ 98.1 | 1896.9$\pm$ 275.8 | 1960.9$\pm$232.7 | 2308.5 $\pm$ 104.4 | 2412.9 $\pm$ 31.6 |
| $\star$non-parallel-MCTS | 1213.5 $\pm$ 169.4 | 1850.1 $\pm$ 281.9 | $-$ | $-$ | $-$ |

Table 5: Zaleplon MPO score (higher the better) with 10 minutes time limit, averaged over 5 runs.

| cores
Method | 4 | 16 | 64 | 256 | 1024 |
|---|---|---|---|---|---|
| MP-MCTS | 0.43 $\pm$0.03 | 0.46$\pm$0.02 | 0.47$\pm$ 0.01 | 0.51$\pm$0.07 | 0.47$\pm$0.01 |

## C.2 WAVELENGTH OPTIMIZATION

It is possible to predict the wavelength of a given molecule using quantum mechanical simulation methods based on Density-Functional Theory (DFT). Following Sumita et al. (Sumita et al., 2018), we apply our MP-MCTS for finding molecules with greater absorption wavelength but limited to 2500 nm.

**Model and dataset.** Our model is based on GRU, which mainly consists of one GRU layer. Input data represents 27 SMILES symbols in one-hot vectors, which represents the symbols appeared in the training dataset. The GRU layer has 27-dim input/256-dim output and the last dense layer has outputs 27 values with softmax. The model was pre-trained using a molecule dataset that contains 13K molecules (only H, O, N, and C elements included) extracted from the PubChemQC database (Nakata & Shimazaki, 2017).

**Calculating wavelength property and reward function.** Following (Sumita et al., 2018), the wavelength property was evaluated by DFT calculations with B3LYP/3-21G* level setting using Gaussian 16 (Frisch et al., 2016). We used one core (for simplicity) for the DFT calculation. The reward function is defined as equation 4. If the generated molecules are valid, then we assign a positive reward within $(-1, 1]$ as shown in equation 4. Negative reward -1 is assigned in case the generated molecule is invalid, DFT fails, or the wavelength is greater than the limit.

$$r = \frac{0.01 \times wavelength}{1 + 0.01 \times |wavelength|} \tag{4}$$

**Optimization of wavelength property.** Table 2 summarizes the optimized wavelength (nm) of MP-MCTS approach for varying number of CPU cores. 3 independent runs and 6 hours time limit were applied to each setting, and the average and standard deviation of the best scores were shown.

## C.3 SIMILARITY SEARCH

To test our MP-MCTS for another type of problem, we ran our algorithm using Zaleplon MPO, one of the Guacamol benchmarks (Brown et al., 2019). Unlike the property based scores (e.g., penalized logP or the wavelength score), Zaleplon MPO aims to find similar but not exactly the same molecules as the given target molecule. Table 5 shows the results for 10 minutes x 5 runs. The best score was 0.659, which we observed in one of the runs for 256 cores. The score is lower than the best score of 0.754 achieved by Graph GA, according to the Guacamol website. Further validation is needed to assess the effectiveness of our method for this type of problem.

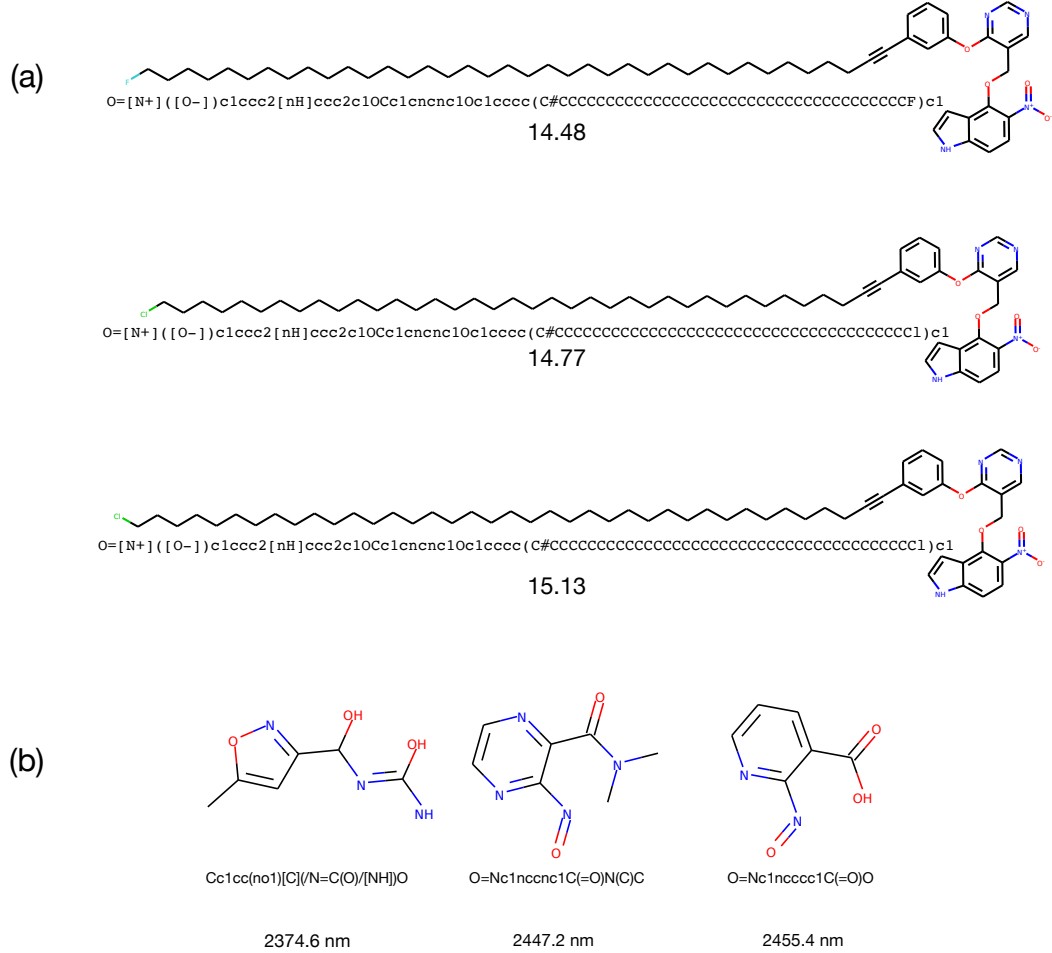

(a)

O=[N+]([O-])c1ccc2[nH]ccc2c1OCc1cncnc1Oc1cccc(C#CCCCCCCCCCCCCCCCCCCCCCCCCCCCCCCCCCCF)c1

14.48

O=[N+]([O-])c1ccc2[nH]ccc2c1OCc1cncnc1Oc1cccc(C#CCCCCCCCCCCCCCCCCCCCCCCCCCCCCCCCCCCCl)c1

14.77

O=[N+]([O-])c1ccc2[nH]ccc2c1OCc1cncnc1Oc1cccc(C#CCCCCCCCCCCCCCCCCCCCCCCCCCCCCCCCCCCCl)c1

15.13

(b)

Cc1cc(no1)[C](/N=C(O)/[NH])O

2374.6 nm

O=Nc1nccnc1C(=O)N(C)C

2447.2 nm

O=Nc1ncccc1C(=O)O

2455.4 nm

Figure 6: Top molecules designed by MP-MCTS (a) Top 3 molecules with highest penalized logP score. (b) Top 3 molecules with highest wavelength/nm property.

## D PSEUDO CODE

The Pseudocode for TDS-UCT, TDS-df-UCT, and MP-MCTS are shown in Algorithm 1, 2, and 3. All of the workers call the function TDS_UCT(), TDS_df_UCT(), or MP_MCTS() respectively, and continue until timeup.

---

**Algorithm 1** Pseudocode for TDS-UCT

---

1: **function** TDS_UCT()
2:     Initialize Hash_Table, Initialize job_queue, N_jobs=3×#workers,
3:     initial_job = [SELECT, root_node]
4:     **if** this worker is the home_worker(root_node) **then**
5:         **for** N_jobs **do**
6:             job_queue.Push(initial_job)
7:         **end for**
8:     **end if**
9:     **while** TimeIsRemaining **do**
10:         Receive all incoming messages and push to job_queue
11:         **if** job_queue.NotEmpty() **then**
12:             (job_type, node) = job_queue.Pop()
13:             **if** job_type == SELECT **then**
14:                 **if** node is in HashTable **then**
15:                     node = LookUpHashTable(node)
16:                     **if** node not yet expanded **then** Expansion(node) **end if**
17:                             ▷ Expand node.children on the second visit
18:                     best_child=Selection(node)
19:                     AddVirtualLoss(node.children[best_child])
20:                     WriteToHashTable(node)
21:                     SendMessage([SELECT, best_child], dest=home_worker(best_child))
22:                 **else**
23:                     Reward=Simulation(node)
24:                     node.Update(Reward)
25:                     WriteToHashTable(node)
26:                     SendMessage([BP, node], dest=home_worker(node.parent))
27:                 **end if**
28:             **else if** job_type == BP **then**
29:                 parent = LookUpHashTable(node.parent) ▷ parent should be in hashtable on BP
30:                 RemoveVirtualLoss(parent.children[node])
31:                 parent.Update(node.reward)
32:                 WriteToHashTable(parent)
33:                 **if** parent != root_node **then**
34:                     SendMessage([BP, parent], dest=home_worker(parent.parent))
35:                 **else**
36:                     best_child=Selection(parent)
37:                     AddVirtualLoss(parent.children[best_child])
38:                     SendMessage([SELECT, best_child], dest=home_worker(best_child))
39:                 **end if**
40:             **end if**
41:         **end if**
42:     **end while**
43: **end function**

---

---

**Algorithm 2** Pseudocode for TDS-df-UCT

---

1: **function** TDS_DF_UCT()
2:     Initialize Hash_Table, Initialize job_queue, N_jobs=3×#workers,
3:     initial_job = [SELECT, root_node, None]
4:     **if** this worker is the home_worker(root_node) **then**
5:         **for** N_jobs **do**
6:             job_queue.Push(initial_job)
7:         **end for**
8:     **end if**
9:     **while** TimeIsRemaining **do**
10:         Receive all incoming messages and push to job_queue
11:         **if** job_queue.NotEmpty() **then**
12:             (job_type, node, ucb_history) = job_queue.Pop()
13:             **if** job_type == SELECT **then**
14:                 **if** node is in HashTable **then**
15:                     node = LookUpHashTable(node)
16:                     **if** node not yet expanded **then** Expansion(node) **end if**
17:                                                     ▷ Expand node.children on the second visit
18:                     best_child=Selection(node)
19:                     AddVirtualLoss(node.children[best_child])
20:                     WriteToHashTable(node)
21:                     ucb_history.Append(node.children)          ▷ ucb_history is only in messages
22:                     SendMessage([SELECT, best_child, ucb_history],
23:                                     dest=home_worker(best_child))
24:                 **else**
25:                     Reward=Simulation(node)
26:                     node.Update(Reward)
27:                     WriteToHashTable(node)
28:                     ucb_history.RemoveBottomRow()          ▷ ucb_history is only in messages
29:                     SendMessage([BP, node, node.ucb_history],
30:                                     dest=home_worker(node.parent))
31:                 **end if**
32:             **else if** job_type == BP **then**
33:                 parent = LookUpHashTable(node.parent) ▷ parent should be in hashtable on BP
34:                 RemoveVirtualLoss(parent.children[node])
35:                 parent.Update(node.reward)
36:                 WriteToHashTable(parent)
37:                 current_best_node = ucb_history.GetCurrentBest()
38:                 ucb_history.RemoveBottomRow()                ▷ ucb_history is only in messages
39:                 **if** parent == current_best_node OR parent == root_node **then**
40:                     best_child=Selection(parent)
41:                     AddVirtualLoss(parent.children[best_child])
42:                     WriteToHashTable(parent)
43:                     ucb_history.Append(parent.children)          ▷ ucb_history is only in messages
44:                     SendMessage([SELECT, best_child, ucb_history],
45:                                     dest=home_worker(best_child))
46:                 **else**
47:                     SendMessage([BP, parent, parent.ucb_history],
48:                                     dest=home_worker(parent.parent))
49:                 **end if**
50:             **end if**
51:         **end if**
52:     **end while**
53: **end function**

---

---

**Algorithm 3** Pseudocode for MP-MCTS

---

1: **function** MP_MCTS()
2:  Initialize Hash_Table, Initialize job_queue, N_jobs=3×#workers,
3:  initial_job = [SELECT, root_node, None]
4:  **if** this worker is the home_worker(root_node) **then**
5:   **for** N_jobs **do**
6:    job_queue.Push(initial_job)
7:   **end for**
8:  **end if**
9:  **while** TimeIsRemaining **do**
10:   Receive all incoming messages and push to job_queue
11:   **if** job_queue.NotEmpty() **then**
12:    (job_type, node, ucb_history) = job_queue.Pop()
13:    **if** job_type == SELECT **then**
14:     **if** node is in HashTable **then**
15:      node = LookUpHashTable(node)
16:      **if** node not yet expanded **then** Expansion(node) **end if**
17:                ▷ Expand node.children on the second visit
18:      best_child=Selection(node)
19:      AddVirtualLoss(node.children[best_child])
20:      node.UpdateUCBHistory(ucb_history)
21:      WriteToHashTable(node)
22:      ucb_history.Append(node.children)
23:      SendMessage([SELECT, best_child, ucb_history],
24:           dest=home_worker(best_child))
25:     **else**
26:      Reward=Simulation(node)
27:      node.Update(Reward)
28:      WriteToHashTable(node)
29:      SendMessage([BP, node, node.ucb_history],
30:           dest=home_worker(node.parent))
31:     **end if**
32:    **else if** job_type == BP **then**
33:     parent = LookUpHashTable(node.parent) ▷ parent should be in hashtable on BP
34:     RemoveVirtualLoss(parent.children[node])
35:     parent.Update(node.reward)
36:     parent.UpdateUCBHistory(ucb_history)
37:     WriteToHashTable(parent)
38:     current_best_node = parent.ucb_history.GetCurrentBest()
39:     **if** parent == current_best_node OR parent == root_node **then**
40:      best_child=Selection(parent)
41:      AddVirtualLoss(parent.children[best_child])
42:      WriteToHashTable(parent)
43:      ucb_history.Append(parent.children)
44:      SendMessage([SELECT, best_child, ucb_history],
45:           dest=home_worker(best_child))
46:     **else**
47:      SendMessage([BP, parent, parent.ucb_history],
48:           dest=home_worker(parent.parent))
49:     **end if**
50:    **end if**
51:   **end if**
52:  **end while**
53: **end function**

---

