# OpenReview forum: "Practical Massively Parallel Monte-Carlo Tree Search Applied to Molecular Design"
_ICLR.cc/2021/Conference — ICLR 2021 Poster_

### Official Review · AnonReviewer3 · 2020-10-26
**Parallelizing MCTS for chemical design; it's domain application is currently under-studied.**

**Rating:** 3
**Confidence:** 3

**Review:**

This paper aims to provide a scalable and competitive molecular design technique based on MCTS. Machine learning guided molecular design is a popular field and many solutions have been developed in the recent years. There is appeal, due to the high dimensionality of search space, to resort to MCTS based approaches that leverage stochastic pruning of the search space as an effective way of making the problem tractable.

Positives:
- The innovations herein are focused on engineering a parallelizable MTCS algorithm that can explore the space much faster (though it requires much higher compute at once).  This can be a useful tool for faster computation and proposal of drug-like molecules

Negatives:
- In principle the methods introduced here have little specialized domain relevance to chemical design. I.e. it's unclear why the paper is focused on chemical design and benchmarks itself there. It seems more appropriate to show that the scalability can be general to any MCTS application. As such, it might be better to apply it in spaces where MCTS is already better developed.

- The paper is light on results and their discussion. Instead of spending as much space talking about how MTCS works, which is well known, the authors are better off showing more metrics related to drug-likeness (e.g. QED) or validity checks.  It is unclear if the solutions proposed are actually sensible which could overstate the performance of the method. Adding long chains of carbon to improve the score is a notorious problem for these design methods, and it does seem to be an issue for the designed molecules presented in the appendix. I'm not a chemist so I cannot validate this molecules by eye, but I am quite concerned about the lack of validity checks.
- Comparison and discussion of more performant methods like Zhou et al 2019 Scientific Reports (optimization of molecules via RL).
- It is also unclear if the scoring method for logP used in this study is the same as the one used by the other methods mentioned. At least it doesn't seem to match the method used in Zhou et al 2019 or others reported there.

Minor comments
Table 1 boldface is confusing, it makes more sense to boldface the best performing models.


------

I think the paper makes some interesting engineering contributions, but the focus on chemical design is somewhat arbitrary, and the authors don't investigate the domain application with enough depth. Especially, I'm quite concerned about validity of the designed molecules, and lack of clarity that all methods reported herein are actually run with the same scoring method (e.g. the numbers aren't simply picked up from elsewhere when the oracle was different).  If the authors are able to provide more reassurance about the validity of the design, and the soundness of the oracle they have chosen to use, I'm happy to improve my score.

---

> ### Author Response · Authors · 2020-11-17
> **Response to Reviewer 3**
>
> Thank you very much for your constructive comments.
>
> **>> In principle the methods introduced here have little specialized domain relevance to chemical design. I.e. it's unclear why the paper is focused on chemical design and benchmarks itself there. It seems more appropriate to show that the scalability can be general to any MCTS application. As such, it might be better to apply it in spaces where MCTS is already better developed.**
>
> A1, To our knowledge, MCTS already achieved successes in several real-world optimization problems, including material science. Also we believe that the use of large scale parallel computational resources would make more sense in these real-world problems than games. The primary purpose of this paper is to show the effectiveness of our parallel MCTS on a recently emerging real-world combinatorial optimization problem, molecular design.
>
> **>> The paper is light on results and their discussion. Instead of spending as much space talking about how MTCS works, which is well known, the authors are better off showing more metrics related to drug-lik eness (e.g. QED) or validity checks. It is unclear if the solutions proposed are actually sensible which could overstate the performance of the method. Adding long chains of carbon to improve the score is a notorious problem for these design methods, and it does seem to be an issue for the designed molecules presented in the appendix. I'm not a chemist so I cannot validate this molecules by eye, but I am quite concerned about the lack of validity checks.**
>
> A2, We believe that we need to explain MCTS because of two reasons. 1, part of the readers will be from chemistry background and would not be familiar with MCTS, 2, it would also help the readers from ML domain understand our parallel method if MCTS is reviewed in the paper.
> For all of our molecules, the validity is confirmed by RDKit. All compared papers, including [Zhou+2019], use the same tool RDKit for the validity check. All generated molecules (not only the top-3 shown in the figure) are “valid” in that regard. This is the same for all compared papers.
> However, we are also fully aware that the penalized logP score is only for benchmark. Our results in the main text focus on penalized logP because other compared methods also used the same penalized logP score (see below A4 for the details of penalized logP).
> Our method and [Zhou et al. 2019] succeeded in exploiting the penalized logP score by finding molecules with long chains. However, we believe that the problem is in the penalized logP score and not the search algorithms. Our results in the appendix shows that if our MP-MCTS is combined with a more reasonable wavelength score, it generates realistic molecules.
>
> **>> Comparison and discussion of more performant methods like Zhou et al 2019 Scientific Reports (optimization of molecules via RL).**
>
> A3, Thank you for your suggestions. We totally agree  doing more benchmarks and comparisons can further strengthen our results. As also suggested by reviewer #5, we will try our method with Guacamol and add the results in the camera ready (if accepted).
>
> **>> It is also unclear if the scoring method for logP used in this study is the same as the one used by the other methods mentioned. At least it doesn't seem to match the method used in Zhou et al 2019 or others reported there.**
>
> A4, We were not aware of this paper by Zhou et al. 2019. Thank you for pointing this out. We will cite the paper and compare the results. Please note that our top 3 scores (15.13, 14.77, 14.48) outperforms the top 3 scores in [Zhou+2019]  (11.84, 11.84, 11.82) by a large margin.
> The logP score is commonly used for benchmark and the definition is exactly the same as other mentioned papers (including [Zhou+2019]). All the papers use the same penalized logP defined by “logP - SA - RP”, calculated by the same tool, RDKit. (SA: synthesized accessibility, RP: ring penalty). (Please refer to [Zhou+2019] page 5, the paragraph starting with “Single property optimization”).
>
> *([Zhou+2019] page 5.)
> "Single property optimization.
> In this task, our goal is to find a molecule that can maximize one selected property. Similar to the setup in previous approaches [13,18], we demonstrated the property optimization task on two targets: penalized logP and Quantitative Estimate of Druglikeness (QED) [28]. LogP is the logarithm of the partition ratio of the solute between octanol and water. Penalized logP [13] is the logP minus the synthetic accessibility (SA) score and the number of long cycles."*
>
> **>> Minor comments Table 1 boldface is confusing, it makes more sense to boldface the best performing models.**
>
> A5, Thank you for pointing this out. we will change to boldface the best performing methods.

---

### Official Review · AnonReviewer1 · 2020-10-29
**An application of a novel parallel MCTS that synthesizes novel molecules and outperforms the state-of-the-art**

**Rating:** 7
**Confidence:** 3

**Review:**

In this paper, the authors propose a novel parallelized implementation of Monte
Carlo Tree Search (MCTS) that is evaluated in a molecule design domain. The MCTS
loop uses the outcomes from prior iterations to guide future expansion of the
search tree --- this complicates attempts to parallelize the algorithm, since
such approaches will need to perform some degree of speculative expansion.
Hash-driven parallelization methods --- where each node in the search tree is
"owned" by a worker --- have been used in the past, together with a "virtual loss"
accounting system and message passing, to successfully parallelize MCTS. By
including a statistics history table in the messages, the TDS-DF-UCT algorithm
has been shown to scale successfully with up to 100 workers by reducing
communication congestion. The authors propose a simple modification to
TDS-DF-UCT called MP-MCTS --- namely, storing the history table in each node,
instead of including it in the messages --- which significantly boosts the
scalability of the algorithm.

In the remainder of the paper, the authors evaluate MP-MCTS in a molecule
synthesis domain, one which has recently drawn significant attention from the ML
community. The goal in this problem is to design novel molecules that maximize a
particular physicochemical property (in this case, a logP measure with
penalties), which is a standard in this area. An RNN pre-trained on a database
of drug molecules is used to generate candidate children at each node in the
search tree, as well as to perform the rollouts. The authors demonstrate that
MP-MCTS displays much better scaling behavior compared to prior parallel MCTS
implementations (TDS-UCT and TDS-DF-UCT) with up to 1024 workers. They also
show striking improvements over prior state-of-the-art methods for molecule
synthesis.

Strengths of the paper:
  + MCTS methods are of broad interest to the AI/ML community and this paper
    offers an interesting application of the algorithm to a new, non-traditional
    domain that has already drawn prior interest from ML researchers (including
    publications at NeurIPS, ICML etc.).
  + The parallel implementation of MCTS described is general and could be
    applied to other problems.
  + The proposed enhancement, while relatively straightforward, nevertheless
    appears to offer significant gains, thus making it easy to adopt.
  + The paper is well-written, organized and easy to follow.

Areas for improvement/questions for the authors:
  - While the results in Table 1 are convincing, I found the results in Table 2
    less so, since computational resources have not been controlled for. While
    there is merit to simply comparing the best molecules found by each method,
    one does have to wonder how well something MolecularRNN might do with the
    equivalent of 1024*10 core minutes.
  - I found that the paper's reasons for MP-MCTS's gains over TDS-DF-UCT
    somewhat unclear. Is it the fact that the messages that need to be passed
    between workers are smaller in size? I also didn't understand what caused
    TDS-DF-UCT to build wider/shallower trees. A slightly more detailed
    discussion of this would be welcome and strengthen the paper.

On balance, I find it noteworthy that such a subtle tweak to an established
algorithm produces such significant improvement in performance on this domain.
While there are some methodological questions and some claims that may need
tempering, I think the results are sufficiently interesting for me to recommend
ACCEPTANCE.

Minor comment:
  - On page 3, the authors write: "and a level d node denotes the d-th symbol
    in a SMILES string". I found this a little confusing at first. I might
    consider rephrasing this along the lines of: "each path through the tree
    represents a left-to-right enumeration of the characters in the SMILES
    string".

---

> ### Author Response · Authors · 2020-11-17
> **Response to Reviewer 1**
>
> Thank you for the constructive feedback.
>
> **>> While the results in Table 1 are convincing, I found the results in Table 2 less so, since computational resources have not been controlled for. While there is merit to simply comparing the best molecules found by each method, one does have to wonder how well something MolecularRNN might do with the equivalent of 1024x10 core minutes.**
>
> A1, The compared papers (except for [Yang et al. 2017]) did not provide the details of the optimization part and the execution time is not known. Table 2 shows the top 3 molecules reported in the papers. For “GRU based”, the result is obtained by 8 hours of computational time using a non-parallel MCTS.
>   According to MolecularRNN [Popova et al. 2019], they repeatedly generated the molecules until the improvement diminished. Therefore we presume they used a sufficiently long time for the optimization part. However, one of the purposes of Table 2 is to show the possibility of further improvement of the complex and better models than the simple GRU model. Our MP-MCTS drastically improved the GRU-based results. We also believe that the combination of our MP-MCTS with other work would be highly promising.
> (Please also refer to our response A1 to reviewer 4.)
>
> **>> I found that the paper's reasons for MP-MCTS's gains over TDS-DF-UCT somewhat unclear. Is it the fact that the messages that need to be passed between workers are smaller in size? I also didn't understand what caused TDS-DF-UCT to build wider/shallower trees. A slightly more detailed discussion of this would be welcome and strengthen the paper.**
>
> A2, Thank you for drawing our attention to this point. For the message size, it is exactly the same as TDS-df-UCT. The difference is in the history table. We also answered a similar question by reviewer 2 about the difference TDS-df-UCT and MP-MCTS. Please kindly refer to our response to A2 reviewer 2 for the detailed explanation.
>
> **>> On page 3, the authors write: "and a level d node denotes the d-th symbol in a SMILES string". I found this a little confusing at first. I might consider rephrasing this along the lines of: "each path through the tree represents a left-to-right enumeration of the characters in the SMILES string".**
>
> A3, Thank you for your suggestion. We consider rephrasing the explanation.

---

### Official Review · AnonReviewer2 · 2020-10-29
**Improves MCTS scalability significantly, nice application and strong results. Algorithm description and analysis could be more detailed.**

**Rating:** 8
**Confidence:** 3

**Review:**

Summary: The paper proposes a new algorithm to scale up parallel MCTS. The proposed method, MP-MCTS is a modified version of previous efforts to parallelize MCTS (TDS-UCT, TDS-df-UCT), all using virtual loss and modern MCTS enhancements (NN-guided selection learned offline). In exchange for small additional memory requirement, MP-MCTS is able to grow significantly deeper trees at much higher levels of parallelism. Overall, the approach is shown to be quite effective at constructing deeper trees efficiently on large MPI clusters and aligns nicely with results in the single-worker MCTS literature (shallow vs deep trees). The experimental section is nicely detailed and shows strong performance gains over the state of the art in molecular design. Overall, this seems to be a very successful scale-up of MCTS, resulting in a new state-of-the-art result in molecular design. The algorithmic description is a bit thin right now. Please consider including a detailed discussion of the pseudo-code.

Strengths:
  + Intuitively clear, reasonably well-written. The illustrative example is very helpful.
  + Proposed algorithm significantly advances the state-of-the-art in distributed, parallel MCTS.
  + Nicely detailed set of experiments showing intuitively clear results.
  + The paper does a good job of making its contributions clear relative to the broader body of work in MCTS and molecular design.

Areas of improvement:
  - The main algorithmic descriptions (Section 3) feel a bit "thin". Details of the algorithmic contributions are difficult to spot on the first read. I had to refer to Algorithms 1 and 2 in the Appendix to understand what was really happening and the pseudo-code seems incomplete. Please consider bringing those 2 algorithms in to the main paper (although space could be tight). Also, pseudo-code for TDS-df-UCT seems to be missing.
  - I'm not sure I fully understood the reasons behind TDS-df-UCT's shallower trees. It seems to have something to do with the search tree being biased towards making local changes (rather than propagating info at every leaf up to the root) but the paper hand-waves a bit about this aspect ("It is presumed that TDS-df-UCT does too few backpropagations"). Figure 5a and 5b illustrate the issues empirically which is nice, but I think a more careful explanation and / or analysis of the root cause would better motivate MP-MCTS.
  - The terminology about workers and home processors feels like it could be simplified a bit. If I understand correctly, each node in the tree is mapped to exactly one worker (MPI process = core). Any message sent to that node will be processed by the assigned worker. Is there more to it than this?

Questions
  * Why does TDS-df-UCT generate shallow trees?
  * What does `node = LookUpHashTable(node)` do?
  * Is the if-else expression in Algorithms 1 and 2 correct? Shouldn't an unsuccessful hash table lookup result in expansion rather than selection? Also, what happens to `new_node`?
  * What exactly is the difference between worker and home processor? Doesn't the same worker handle every message sent to the tree nodes it's been assigned to?
  * Will you be releasing source code? I didn't see any links and the pseudo-code is incomplete.


UPDATE: I thank the authors for their detailed feedback and updated paper. I'm a bit more in favor of acceptance of the paper now.

---

> ### Author Response · Authors · 2020-11-17
> **Response to Reviewer 2**
>
> Thank you very much for the useful suggestions and we appreciate the detailed comments about our pseudo code. We sincerely apologize for the incorrectness of the pseudo code. It differs from our actual code, which is included in the supplementary material.
>
> **>> The main algorithmic descriptions (Section 3) feel a bit "thin". Details of the algorithmic contributions are difficult to spot on the first read. I had to refer to Algorithms 1 and 2 in the Appendix to understand what was really happening and the pseudo-code seems incomplete. Please consider bringing those 2 algorithms in to the main paper (although space could be tight). Also, pseudo-code for TDS-df-UCT seems to be missing.**
>
> A1, We apologize for the mistakes in our pseudo code. We will improve the pseudo-code and also add the pseudo-code for TDS-df-UCT.
>
> **>> I'm not sure I fully understood the reasons behind TDS-df-UCT's shallower trees. It seems to have something to do with the search tree being biased towards making local changes (rather than propagating info at every leaf up to the root) but the paper hand-waves a bit about this aspect ("It is presumed that TDS-df-UCT does too few backpropagations"). Figure 5a and 5b illustrate the issues empirically which is nice, but I think a more careful explanation and / or analysis of the root cause would better motivate MP-MCTS.**
>
> A2, The difference between TDS-df-UCT and MP-MCTS is that MP-MCTS maintains a history table that is stored in each node, while tds-df-uct does not store the history table in the node. For MP-MCTS, the history table in each node is updated when a job visits the node. On the other hand, TDS-df-UCT carries the history table only in the messages, and it is needed to return to the root node to update all the entries of the history table. Therefore, the jobs of TDS-df-UCT relies on more outdated information to decide whether they should backpropagate or not. Because of this, the jobs tend to stay longer in the suboptimal subtrees and the tree will become wider and shallower.
> The shallower tree also affects the average time needed for rollouts. Because we limit the maximum length of SMILES (to 81 in our case), the rollout steps would be fewer for deeper trees. This effect makes it possible for MP-MCTS to perform more rollouts, and therefore further emphasizes the difference between TDS-df-UCT and MP-MCTS.
>
> **>> The terminology about workers and home processors feels like it could be simplified a bit. If I understand correctly, each node in the tree is mapped to exactly one worker (MPI process = core). Any message sent to that node will be processed by the assigned worker. Is there more to it than this?**
>
> A3, Your understanding is perfectly correct. The “home processor” of a node is the worker which holds the node in its hash table. We used the term “home processor” because it was used in existing papers, but we consider changing the term to “home worker” to avoid confusion.
>
> **>> Why does TDS-df-UCT generate shallow trees?**
>
> A4, Please refer to our comment A2.
>
> **>> What does node = LookUpHashTable(node) do?**
>
> A5, In our implementation, if the node exists in the hash table, LookUpHashTable returns the node, otherwise it returns None. We apologize for not explaining the details. We will revise our pseudo code.
>
> **>> Is the if-else expression in Algorithms 1 and 2 correct? Shouldn't an unsuccessful hash table lookup result in expansion rather than selection? Also, what happens to new_node?**
>
> A6, Thank you very much for pointing this out. We apologize for this incorrectness and will correct this mistake. Yes, it was the opposite. An unsuccessful hash table lookup indicates that the node is a leaf node and a successful lookup means it was an inner node. We will fix our pseudo code accordingly.
>
> **>> What exactly is the difference between worker and home processor? Doesn't the same worker handle every message sent to the tree nodes it's been assigned to?**
>
> A7, Please refer to our comment above.
>
> **>> Will you be releasing source code? I didn't see any links and the pseudo-code is incomplete.**
>
> A8, The code is included in the supplementary material. Also, we plan to make the code public after this paper gets accepted at a conference.

---

### Official Review · AnonReviewer4 · 2020-11-02
**Significant speedup for MCTS algorithms for the Molecular Design problems. The technical contribution is rather limited.**

**Rating:** 5
**Confidence:** 2

**Review:**

In this work the authors apply a distributed parallel Monte-Carlo Tree Search (MCTS) algorithm to the Molecular Design problem. The goal of this paper is to speed-up the computation needed by an MCTS algorithm using parallelization over multiple machines. The authors show how to modify a previously successful parallel MCTS algorithm (namely, TDS-DF-UCT) to address its shortcoming that leads to shallow and wide trees. The modification that is performed changes the way that the history on the nodes are stored and propagated, which leads the algorithm to construct trees with more similar characteristics with those constructed by a sequential algorithm that is allowed to use time equal to the number of processors times the time required by the parallel implementation.

The authors then apply their algorithm to the molecular design problem using a pre-trained GRU-based model for rollout and expansion, and compared the quality of the method with non-parallel implementations (that are allowed to use time equal to the time of the parallel algorithm times the number of CPUs used; the goal is to obtain solutions with quality close to the performance of such algorithms), and to existing parallel implementations of MCTS algorithms (that lead to non-ideal MCTS tree structures). The experiments show that, in terms of quality, the proposed parallel MCTS algorithm outperforms the rest of the parallel MCTS algorithms, given the same amount of time, and performs closer to the non-parallel implementation which takes much more time.

Finally, the authors compare their method, in terms of quality, to state-of-the-art algorithms for the problem and show that their method outperforms the rest of the methods significantly.

PROS:
+ The experimental results show that the proposed algorithms coupled with the necessary model achieve very good parallelization
+ The proposed algorithm outperforms the state-of-the-art approaches on an application to the Molecular Design problem
+ This is the first work that applied the distributed parallel MCTS algorithms for the Molecular Design problem

CONS:
- The technical contribution of the paper seems rather incremental to previous work on parallel MCTS algorithms

The proposed method shows a significant speedup for the molecular design problem, and for nearly-linear speedup similar solution quality is achieved. Although I'm not an expert in Molecular Design, it seems to me that this work should also be of interest to people in Computational Chemistry. On the other hand, the new methodology is not tight to the specific application and might find applications to other problems where MCTS algorithms were found to be successful.
My main complaint is the incremental technical contribution of the paper, which is limited to adapting the history message passing and storing in the parallel MCTS method proposed by (Yoshizoe et al 2011). On the other hand, it is evident that this small change to the algorithm is able to achieve a much more desirable MCTS tree structure and performance.

Overall, I tend to be positive about the paper, given it's claimed significant contribution in the application to the Molecular Design problem.

Small things:
* In Table 2, how long did you let the competitor algorithms run? Was it again 10 minutes?
* Is the massively parallel terminology widely used for distributed parallel algorithms? This conflicts with the terminology for the Massively Parallel Computation model, which is an abstraction of MapReduce, Hadoop, and other systems. I suggest using "Distributed Parallel".
* "the the" in page 3.

---

> ### Author Response · Authors · 2020-11-17
> **Response to Reviewer 4**
>
> Thank you for the thorough review.
>
> **>> In Table 2, how long did you let the competitor algorithms run? Was it again 10 minutes?**
>
> A1, We are showing the best results for the competitor algorithms provided in the cited papers. It seems it is not common to report the execution time for the search (or optimization) part, in this domain.  Among the competitors, only GRU-based [Yang et al. 2017] reported the execution time for the search part, which is 8 hours. Other competitors (including the newly added [Zhou+2019] based on reviewer 3’s comment) did not explain the execution time. We presume they spent much longer than 10 minutes, but we can only guess. FYI: JT-VAE bayesian optimization, GCPN, MolecularRNN used reinforcement learning, Mol-CycleGAN repeatedly generated the molecules using the trained generator.
>   Table 2 shows mainly two things. One is our higher score. Another is the possible improvement to existing models using parallel MCTS. Our MP-MCTS used the same model as GRU-based and the score was greatly improved, which shows the possibility of improving the results of other more complex models.
>   We will revise our paper to remove the vagueness.
> (Please also refer to our response A1 to reviewer 1.)
>
> **>> Is the massively parallel terminology widely used for distributed parallel algorithms? This conflicts with the terminology for the Massively Parallel Computation model, which is an abstraction of MapReduce, Hadoop, and other systems. I suggest using "Distributed Parallel".**
>
> A2, Thank you for the suggestion. We agree that it is meaningful to emphasize that our method works for Distributed Memory environments. However, we also want to emphasize that our method is parallelized at a large scale. In the HPC (High Performance Computing) domain, “massively parallel” is a commonly used phrase which simply means something parallelized at a massive scale (regardless of hardware or software). We will consider to improve the title and/or the abstract based on your suggestion.

---

### Official Review · AnonReviewer5 · 2020-11-08
**new approach to MCTS parallelization seems to be useful**

**Rating:** 7
**Confidence:** 4

**Review:**

The authors present an approach to parallelize MCTS on a large number of processors. Out of potentially many different applications, they chose to use the domain of molecular design, and show improved performance over several baselines, including some baselines using non-parallel MCTS. Impressively, this is achieved in very little wall time.


Strengths:
-  Very simple idea (this is good), straightforward execution, no faff
- The approach is non domain specific, and could be applied in many different areas
- Good empirical results


Weaknesses:
- Not very fancy from the ML side
- The validation is weak, and to some extent outdated. I understand that drug design application is only an example, however, nevertheless I would strongly suggest that the authors run the more challenging guacamol benchmark, since the benchmarks the authors use are know to be very easy (see https://github.com/BenevolentAI/guacamol / https://pubs.acs.org/doi/abs/10.1021/acs.jcim.8b00839 )

Question:
- How much tuning of the size of the virtual loss is required to optimally use the provided CPUs?
- By using the virtual loss, do we lose some of the theoretical guarantees of UCT?

All in all, I think this paper could in my opinion be accepted at ICLR if the authors commit to running the harder guacamol benchmarks for the camera ready version.

---

> ### Author Response · Authors · 2020-11-17
> **Response to Reviewer 5**
>
> Thank you for the constructive suggestions.
>
> **>> The validation is weak, and to some extent outdated. I understand that drug design application is only an example, however, nevertheless I would strongly suggest that the authors run the more challenging guacamol benchmark, since the benchmarks the authors use are know to be very easy (see https://github.com/BenevolentAI/guacamol / https://pubs.acs.org/doi/abs/10.1021/acs.jcim.8b00839 )**
>
> A1, We will run the Guacamol benchmark as suggested, and we think this is very important for further evaluating the performance of our MP-MCTS. We plan to 1) train an RNN model using the Guacamol training data. 2) use one of the goal-oriented benchmark scores instead of the penalized logP score. We appreciate it if you could show us the recommended settings (scores) when using the Guacamol benchmark.
>
> **>> How much tuning of the size of the virtual loss is required to optimally use the provided CPUs?**
>
> A2, We did not do any tuning of the virtual loss value. We used the most straightforward (pessimistic) approach, which assumes zero rewards. It is possible that the results could be improved by tuning the virtual loss size.
>
> **>> By using the virtual loss, do we lose some of the theoretical guarantees of UCT?**
>
> A3, There is no published result about the theoretical guarantee of virtual loss based parallel UCT. Many applications (including parallel Go programs) started to use virtual loss based parallel UCT in 2008. However, the proof still remains an open problem.  One of the assumptions required by the proof of UCT is the 1-Lipschitz condition. The virtual Loss itself will not violate the assumption, but TDS-df-UCT and MP-MCTS break the assumptions because they increment the visit count and reward at once (for multiple simulations). It is possible to guarantee the convergence if we increase the exploration constant, but that would make the tree wider (and shallower) and degrade the overall performance. Formal theoretical analysis remains future work.

---

> > ### Comment · AnonReviewer5 · 2020-11-17
> > **Re:A1**
> >
> > Re A1:thank you for your answer.
> > I‘ve checked the guacamol repository, and it seems that there is even pretrained RNN available which you could reuse ( linked under guacamol_baselines) which may make this task even easier.

---

### Decision · Program_Chairs · 2021-01-07
**Final Decision**

**Decision:**

Accept (Poster)

**Comment:**

I think this is a very solid and good work in the topic of "Practical Massive Parallel MCTS."   I think it will be good to open up perspectives among ICLR's audience going beyond just Deep Learning and Machine Learning. I also noted a lot of positive comments during the evaluation and discussion period.

Still, it was a borderline case and not an easy decision (primarily because of the concerns raised by R3 towards the end of the discussion period). In the end the program committee decided that the paper does meet the bar.  We think that the work is interesting and original, though not without weaknesses.